# SCAN: Bootstrapping Contrastive Pre-training for Data Efficiency

## Abstract

While contrastive pre-training is widely employed, its data efficiency problem has remained relatively under-explored thus far. Existing methods often rely on static coreset selection algorithms to pre-identify important data for training. However, this static nature renders them unable to dynamically track the data usefulness throughout pre-training, leading to subpar pre-trained models. To address this challenge, our paper introduces a novel dynamic bootstrapping dataset pruning method. It involves pruning data preparation followed by dataset mutation operations, both of which undergo iterative and dynamic updates. We apply this method to two prevalent contrastive pre-training frameworks: **CLIP** and **MoCo**, representing vision-language and vision-centric domains, respectively. In particular, we individually pre-train seven CLIP models on two large-scale image-text pair datasets, and two MoCo models on the ImageNet dataset, resulting in a total of 16 pre-trained models. With a data pruning rate of 30-35% across all 16 models, our method exhibits only marginal performance degradation (less than **1%** on average) compared to corresponding models trained on the full dataset counterparts across various downstream datasets, and also surpasses several baselines with a large performance margin. Additionally, the byproduct from our method, *i.e.*, coresets derived from the original datasets after pre-training, also demonstrates significant superiority in terms of downstream performance over other static coreset selection approaches. We include the code in the supplementary material to facilitate the reproduction of our experimental results.

## 1 Introduction

Large models are heavily data-driven, particularly in the realm of pre-training (Chen et al., 2020c; 2021; Radford et al., 2021). This paradigm has been widely underpinned by the scaling law (Hoffmann et al., 2022; Kaplan et al., 2020), which suggests that more data often lead to reduced generalization errors. However, using large quantities of data frequently results in a notable increase in carbon footprints. Addressing this pressing issue requires substantial efforts to optimize the data efficiency.

This paper delves into the data efficiency problem for contrastive pre-training. Despite the pervasiveness of contrastive pre-training across both vision-centric (Chen et al., 2020c; 2021) and vision-language (Jia et al., 2021; Radford et al., 2021) domains, nevertheless, the data efficiency issue has received scant attention in the existing literature. We attribute the reason for this fact to two challenges. **I**- Absence of reliable labels for self-supervised learning objectives. Unlike in supervised learning, where explicit labels aid in class prediction, self-supervised learning in contrastive pre-training operates without such guidance, making it unable to estimate the class probability of data samples such as EL2N (Paul et al., 2021). **II**- Extensive data scale due to easy accessibility, *e.g.*, the interleaved image-text data from the web (Radford et al., 2021). Current datasets usually comprise millions (Changpinyo et al., 2021; Sharma et al., 2018) or even billions of samples (Schuhmann et al., 2022). It is thus intractable in time for methods employing gradients (Paul et al., 2021) or second derivative (Influence Functions) (Koh & Liang, 2017) to evaluate data usefulness individually. Recent approaches in the vision-language area have resorted to coreset selection algorithms (Mirzasoleiman et al., 2020) for a reduced pre-training dataset beforehand (Abbas et al., 2023; Maharana et al., 2024; Mahmoud et al., 2024; Wang et al., 2023; Webster et al., 2023). The crux of these methods lies in the semantic match/duplication that is quantified by some proxy metrics like CLIP matching

score (Radford et al., 2021). Consequently, a subset, namely a coreset, of the original dataset is filtered for pre-training from scratch.

Our motivation for this work is inspired by the advancement in dynamic sparse training (DST) (Nowak et al., 2023; Yuan et al., 2021; Zhang et al., 2024), which dynamically prunes less influential learnable weights from models. Compared to its static sparse training counterparts (Lee et al., 2019; Tanaka et al., 2020), DST demonstrates notable strengths in performance, robustness, and model compression without the need for over-parameterization (Liu et al., 2021a; Nowak et al., 2023). Intriguingly, we recognize that recent coreset selection algorithms predominantly adhere to the static approach, akin to the fixed weight masks employed in static sparse networks (Lee et al., 2019). As a result, we argue that these coreset-based dataset pruning methods are subject to similar limitations as previous static sparse training ones, albeit with a shift in application scope from learnable weights to individual data samples. Given this context, approaching the dataset pruning challenge can be easily decomposed into two sub-problems: **1**)- **metric identification** and **2**)- **pruning strategy design**. Specifically, the proxy metric should meet several conditions: dynamic adaptability, quick obtainability (with minimal additional cost), and reflecting the learning status of each sample. Regarding the pruning strategy, we introduce a novel data bootstrapping algorithm named **SCAN**. Instead of employing a consistent pruning ratio throughout training, our SCAN approach identifies and eliminates data from less important subsets in a bootstrapping manner. These two operations from our SCAN method are performed iteratively for stable pre-training.

We validate the effectiveness of the proposed method with widely used contrastive pre-training frameworks in both vision-language (**CLIP** (Ilharco et al., 2021; Radford et al., 2021)) and vision-centric (**MoCo** (Chen et al., 2020c; 2021)) domains. The pre-training datasets for CLIP include CC3M (Sharma et al., 2018), MSCOCO (Lin et al., 2014), SBU-Captions (Ordonez et al., 2011), and CC12M (Changpinyo et al., 2021), forming two groups of datasets with different scales. On the other hand, we pre-train MoCo models using the ImageNet Dataset (Deng et al., 2009). Moreover, we employ various downstream datasets, including ImageNet (Deng et al., 2009), CIFAR-10, and CIFAR-100 (Krizhevsky et al., 2009), along with out-of-distribution datasets such as ImageNet-R (Hendrycks et al., 2021) and ImageNet V2 (Recht et al., 2019). Our evaluation protocols encompass full fine-tuning, linear probing, and zero-shot testing on ImageNet. Within the CLIP framework, we evaluate seven models covering ResNet (He et al., 2016),

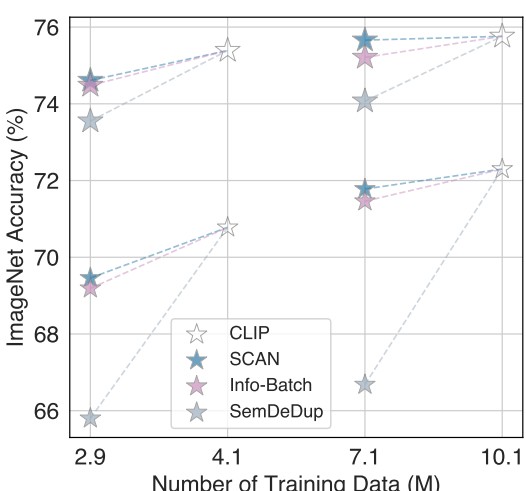

Figure 1: The interplay between training data size and model downstream performance of base model CLIP, our method SCAN, and two SoTA baselines.

ViT (Dosovitskiy et al., 2021), and Swin Transformer (Liu et al., 2021b). As for MoCo, due to resource constraints, we use two popular ViT models (Touvron et al., 2021) in the experiments. Our experimental results, partially depicted in Fig. 1, exhibit that SCAN achieves a significant trade-off between training data size and downstream model performance as compared to several baselines (Abbas et al., 2023; Qin et al., 2024; Yang et al., 2023). We include the code in the supplementary material for the reproduction of our results.

To the best of our knowledge, we are the first to comprehensively study the data efficiency problem within the context of contrastive pre-training. Our work not only introduces an effective bootstrapping approach but also is able to produce a static coreset (a smaller dataset) that outperforms other static coreset selection methods (Abbas et al., 2023; Yang et al., 2023) by a large performance margin on diverse downstream image classification datasets. These contributions enable our work to hold a positive promise for the efficient utilization of data in contrastive pre-training, thereby potentially reducing more computational overhead and carbon footprints.

## 2 RELATED WORK

**Dataset Pruning and Distillation** represent two common approaches to enhancing dataset efficiency during training. The former aims to synthesize a smaller dataset that achieves test performance similar to a full dataset when using the same model (Chen et al., 2024; Du et al., 2023; Shang et al., 2023; Sun et al., 2024). Recent advancements in this area have leveraged techniques such as mutual information (Shang et al., 2023), frequency domain transformation (Shin et al., 2023), and multi-stage generation (Chen et al., 2024) to craft datasets that exhibit enhanced performance. Two notable limitations are inherent in these methods: I) The generalization capability is significantly constrained due to the reliance on distillation from a specific dataset and model, *e.g.*, the use of ResNet (He et al., 2016). II) The dataset sizes utilized for comparison are frequently confined to small-scale datasets such as CIFAR (Krizhevsky et al., 2009) and Tiny-ImageNet (Le & Yang, 2015). In contrast, dataset pruning is employed to directly filter a smaller subset from the original dataset (Li et al., 2024; Sorscher et al., 2022). Typically, previous methods involve initially learning an indication score, which serves as a basis for identifying and subsequently removing data samples falling below or above a certain threshold. For image classification tasks, prevailing methods often utilize gradient (Paul et al., 2021), loss value (Qin et al., 2024), and second derivative (Koh & Liang, 2017) as the indicator. More recently, efforts have emerged focusing on pruning vision-language pre-trained datasets (Li et al., 2023a;b; Wang et al., 2023; Xu et al., 2023; 2024). The key idea is to construct a coreset by identifying the semantic mismatch/duplication, a process facilitated by often using a pre-trained CLIP model (Beaumont, 2022; Radford et al., 2021).

**Contrastive Pre-Training** has garnered wide attention as a technique for pre-training versatile models applicable to a range of downstream tasks (Gao et al., 2021). Its fundamental principle involves bringing the embeddings of positive pairs closer while simultaneously pushing away negative ones. Traditional approaches within the vision-centric domain often build positive samples by leveraging alternative views of the anchor sample, as exemplified by approaches like SimCLR (Chen et al., 2020a;b) and MoCo (Chen et al., 2020c; 2021). Benefiting from the advancement of transformer architectures (Dosovitskiy et al., 2021; Vaswani et al., 2017), recent endeavors have shifted towards patch masking followed by subsequent recovery (Bao et al., 2022; Caron et al., 2021; He et al., 2022). Contrastive learning has achieved notable success in vision-language per-training as well (Jia et al., 2021; Radford et al., 2021). The alignment of modalities has propelled significant advancements in downstream multi-modal tasks, including visual question answering (Antol et al., 2015; Zhou et al., 2022) and cross-modal retrieval (Bowyer & Flynn, 2000; Lin et al., 2014). Our study specifically targets the data efficiency challenge within CLIP and MoCo, which serve as prominent representatives of vision-language and vision-centric doamins, respectively.

**Dynamic Sparse Training (DST).** Unlike earlier static sparse training methodologies (Lee et al., 2019; Tanaka et al., 2020), DST learns a sparse neural network by pruning weights and growing them back throughout the training process. The weight importance is typically quantified using metrics such as magnitude (Evci et al., 2020; Mocanu et al., 2018), gradients (Yuan et al., 2021), or sensitivity (Mozer & Smolensky, 1989). The demonstrated superiority of DST over its static counterpart inspires us to design a dynamic approach for dataset pruning, especially considering that recent coreset-based methods predominantly adhere to fixed pruning strategies. Furthermore, the well-developed DST methods provide additional hints for devising our dataset pruning strategy.

## 3 METHOD

### 3.1 BACKGROUND OF CONTRASTIVE PRE-TRAINING

Contrastive pre-training necessitates the utilization of both positive and negative pairs of samples, be it alternative views of an image (Chen et al., 2020c; 2021) or combinations of image and text (Jia et al., 2021; Radford et al., 2021). Its objective is to bring positive pairs closer in the embedding space while pushing negative ones away. At its core is the InfoNCE loss (van den Oord et al., 2018), defined as,

$$\mathcal{L}_{f \to g} = -\frac{1}{|\mathcal{D}_t|} \sum_{i=1}^{|\mathcal{D}_t|} \log \frac{\exp(f(I_i)^T g(T_i)/\tau)}{\sum_{j=1}^{|\mathcal{D}_t|} \exp(f(I_i)^T g(T_j)/\tau)}, \tag{1}$$

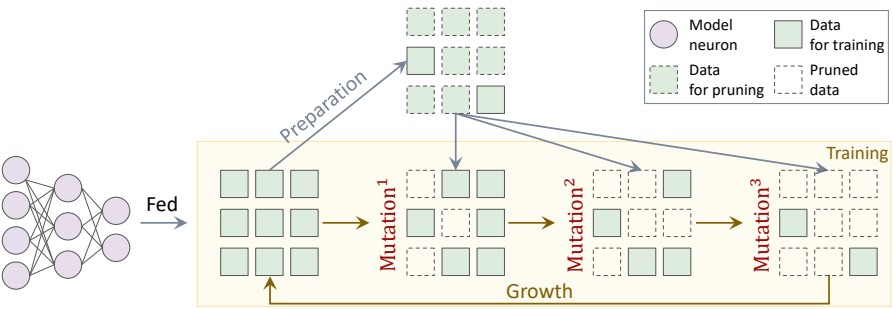

Figure 2: Overall pipeline of the proposed SCAN method. We begin by identifying a substantial portion of data samples as pruning candidates. Subsequently, a subset of these candidates is employed for pruning based on varying mutation ratios that are gradually increased (bootstrapping). After growing back to the original full dataset, the above two operations are iterated for another round.

where $\tau$ is a learnable temperature, $I_i$ and $T_j$ respectively denote the sampled image and text from the batched examples $\mathcal{D}_t$; while $f$ and $g$ represent the image and text encoders, respectively. Likewise, we can obtain the loss from the other direction $\mathcal{L}_{g \to f}$ as well. The overall training loss is then computed as the mean of $\mathcal{L}_{f \to g} + \mathcal{L}_{g \to f}$. Without loss of generality, we take the contrastive learning utilized in CLIP as an example (Radford et al., 2021). The application in other approaches such as MoCo (Chen et al., 2021) can be straightforwardly extrapolated.

**Motivation.** Contrastive pre-training often demands large-scale data to learn a versatile model. For instance, the original CLIP pre-training uses 400M image-text pairs sourced from the web (Radford et al., 2021), while recent studies push these limits to datasets containing billions of samples (Schuhmann et al., 2022; Yu et al., 2022). Accordingly, the introduced footprint and storage cost present significant challenges for researchers. To address this issue, we propose a novel boot**S**trapping **C**ontr**A**stive Pre-trai**N**ing method (**SCAN**), to dynamically, efficiently, and effectively leverage smaller dataset for pre-training[1].

## 3.2 SCAN OVERVIEW

Our proposed SCAN method involves a two-step operation. Specifically, our **first** step entails identifying a proxy metric that is dynamically adaptable, easily obtainable, and capable of tracing the learning status of each sample. We abandon the use of gradients as done in related domains (Paul et al., 2021), given the substantially increased compute overhead incurred for individual samples[2]. Instead, we opt for the loss value as the reliable indicator as it meets the above conditions. Under this context, we disentangle the loss in Eqn. 1 to obtain a loss set $\bar{\mathcal{L}}_{f \to g} = \{ \bar{\mathcal{L}}_{f \to g}^{(1)}, \bar{\mathcal{L}}_{f \to g}^{(2)}, ..., \bar{\mathcal{L}}_{f \to g}^{(|\mathcal{D}_t|)} \}$, with each element corresponding to the loss value of one data sample.

The **second** core step is to determine the pruning strategy. Addressing sparsity within the dataset pruning study presents a substantial challenge. To approach this, we propose a pruned data preparation and then a dataset mutation approach. Specifically, in the pruned data preparation stage, candidate data samples that are potentially less important are selected. Thereafter, in the dataset mutation stage, samples are gradually bootstrapped for pruning across epochs. These two stages iterate through several rounds until the completion of pre-training.

## 3.3 BOOTSTRAPPING DATASET PRUNING

Unlike conventional approaches that use the full dataset for pre-training, we vary the training data size for each epoch. As illustrated in Fig. 2, an example case utilizes 1.0, ⅘, ⅕, and ⅖ of the full dataset for four consecutive training epochs, resulting in an average dataset pruning ratio of ∼40%. More details regarding the algorithm can be kindly found at Appendix A.

---

[1]The name itself reflects our method's capability to *scan* all data samples, identifying those that should be eliminated from further pre-training.

[2]The data size in contrastive pre-training is notably larger than in other related domains.

### 3.3.1 PRUNING DATA PREPARATION

We opt to utilize the loss values of in-batch samples rather than those from the entire dataset for candidate selection. This decision is based on two facts: 1) Comparing InfoNCE losses across batches holds less significance compared to supervised learning, as the instance loss varies drastically with respect to the randomly selected batched samples. 2) Saving the losses for the entire dataset incurs more computational storage compared to in-batch ones.

Additionally, existing coreset selection approaches (Hessel et al., 2021; Schuhmann et al., 2021) from the vision-language domain typically focus only on pruning *ill-matched* samples. These are characterized by image-text pairs indicating less semantic alignment, where, for example, the text inadequately describes the content of the paired image. However, we posit that beyond these ill-matched samples, there also exist samples that are *redundant*. These redundant samples are effectively memorized during the early stages of training and are less likely to be forgotten with further training iterations (Feldman & Zhang, 2020). In view of this, we can safely eliminate these redundant data as training progresses.

To operationalize the above ideas, given a pruning ratio $\rho$, we separately identify the *ill-matched* and *redundant* set of data using:

$$
\begin{cases}
\mathcal{D}_t^{red} & = \mathcal{D}_{t:i}, \quad i \in {}_{\prec_\rho}\bar{\mathcal{L}}_{f \to g}, \\
\mathcal{D}_t^{ill} & = \mathcal{D}_{t:j}, \quad j \in {}_{\succ_\rho}\bar{\mathcal{L}}_{f \to g},
\end{cases}
\tag{2}
$$

where ${}_{\prec_\rho}\bar{\mathcal{L}}_{f \to g}$ denotes the indices of the $\rho$ smallest values of $\bar{\mathcal{L}}_{f \to g}$ (the loss set before loss summation and backpropagation). We then obtain the *redundant* subset $\mathcal{D}_t^{red}$ by selecting data samples according to these indices from the original full in-batch set $\mathcal{D}_t$. This approach is driven by the intuition that small losses often denote effective data memorization by the given model. On the other hand, we can also identify the *ill-matched* subject $\mathcal{D}_t^{ill}$ from the $\rho$ largest loss values using ${}_{\succ_\rho}\bar{\mathcal{L}}_{f \to g}$. This is because large losses are usually associated with small cosine similarities (Hessel et al., 2021), indicating a *poor match* between image and text pairs. The final candidate subset is formed by the union of these two: $\mathcal{D}_t^{'} = \mathcal{D}_t^{red} | \mathcal{D}_t^{ill}$.

Thereafter, we merge the subset intersection from $\mathcal{L}_{f \to g}$ and $\mathcal{L}_{g \to f}$. In total, we obtain $2\rho|\mathcal{D}_t|$ candidates for the current batched samples, which amounts to twice the size of the expected pruned data, as will be explained in the next section. At last, we iterate through all training data to have the final candidate pruning subset $\mathcal{D}^{'}$.

**Preparation Warm-up.** It is intuitive that the model's learning capability may exhibit instability during the early training iterations. To mitigate this issue, we design a warm-up strategy (Gotmare et al., 2019), wherein the full dataset is utilized for training during the first several epochs. We track the average epoch-wise loss to determine the optimal timing for initiating pruning. Specifically, we calculate the difference between the loss from the previous epoch $\mathcal{L}_{pre}^{'}$ and the current epoch $\mathcal{L}_{cur}^{'}$, and compare it against a pre-defined threshold value $T_{td}$. If the condition $(\mathcal{L}_{pre}^{'} - \mathcal{L}_{cur}^{'})/(\mathcal{L}_{pre}^{'} + \epsilon) \geq T_{td}$ holds true, where $\epsilon$ is infinitesimal and is introduced to prevent overflow, it indicates relative stability in the pre-training process. Consequently, we can then start the *pruning data preparation* from this pre-training epoch.

### 3.3.2 DATASET MUTATION

Rather than employing a static pruning ratio consistently throughout training, we advocate for a bootstrapping dataset mutation approach. The benefits of this methodology are shown in Sect. 4.2. Additionally, we refrain from performing the **pruning data preparation** solely once, as further training iterations may alter the matching and redundancy characteristics of data samples.

Specifically, we regenerate the candidate pruning data from scratch every $(\tau_{cos} + 1)$ epochs as detailed in Sect. 3.3.1. Subsequently, we adapt the cosine annealing strategy (Loshchilov & Hutter, 2017) to determine the current pruning ratio $\rho_{cur}$ using:

$$
\rho_{cur} = \frac{1}{2}(1 + \cos((\tau_{cos} - (\tau_{cur} \bmod (\tau_{cos} + 1)))\frac{\pi}{\tau_{cos}})).
\tag{3}
$$

It can be easily seen that the pruning ratio $\rho_{cur}$ gradually and periodically increases with larger training epoch $\tau_{cur}$. We can then randomly select $\rho_{cur}|\mathcal{D}^{'}|$ samples from $\mathcal{D}^{'}$ for pruning $- \mathcal{D}_{\rho}^{'}$.

Table 1: Performance comparison of CLIP models on the **CC12M+** pre-trained datasets. CLIP utilizes **10.1M** pre-trained data samples, while the remaining models use **7.1M**. The best results (excluding the original CLIP model) are highlighted in **bold**. A dash (-) indicates the collapse of pre-training, resulting in impaired evaluation of downstream tasks.

| Architecture | Method | IN Zero-Shot | | CIFAR10 | CIFAR100 | IN | IN-V2 | IN-R |
| --- | --- | --- | --- | --- | --- | --- | --- | --- |
| | | Top-1 | Top-5 | | | | | |
| RN101 | CLIP | 18.78 | 41.14 | 95.96 | 82.13 | 75.76 | 64.31 | 40.57 |
| | Random | 14.05 | 30.60 | 95.02 | 78.34 | 73.99 | 60.27 | 36.13 |
| | SemDeDup | 13.26 | 29.70 | 95.07 | 78.77 | 74.24 | 62.16 | 37.65 |
| | D-Pruning | 12.59 | 28.62 | 94.94 | 78.89 | 74.07 | 61.30 | 37.07 |
| | Info-Batch | 21.60 | 41.11 | 96.04 | 81.60 | 75.21 | 63.27 | 39.34 |
| | SCAN | **23.10** | **47.52** | **96.08** | **82.28** | **75.66** | **63.75** | **40.10** |
| ViT-B/32 | CLIP | 24.62 | 49.10 | 95.62 | 82.11 | 63.40 | 49.97 | 31.09 |
| | Random | 09.12 | 21.09 | 90.13 | 69.98 | 51.99 | 41.01 | 20.08 |
| | SemDeDup | 06.47 | 16.71 | 90.83 | 70.03 | 52.21 | 39.75 | 20.99 |
| | D-Pruning | 06.27 | 15.88 | 90.11 | 69.69 | 51.72 | 38.66 | 20.42 |
| | Info-Batch | - | - | - | - | - | - | - |
| | SCAN | **26.12** | **50.67** | **95.41** | **81.16** | **61.55** | **48.73** | **29.23** |
| ViT-B/16 | CLIP | 23.43 | 47.71 | 96.76 | 84.25 | 72.30 | 59.39 | 33.50 |
| | Random | 14.45 | 32.41 | 94.35 | 76.45 | 67.09 | 54.38 | 27.07 |
| | SemDeDup | 11.58 | 26.01 | 94.18 | 76.71 | 67.22 | 53.78 | 27.19 |
| | D-Pruning | 10.00 | 23.72 | 93.82 | 75.96 | 66.68 | 53.13 | 26.23 |
| | Info-Batch | 22.12 | 42.30 | 96.03 | 81.69 | 71.46 | 56.35 | 31.13 |
| | SCAN | **24.71** | **49.12** | **96.13** | **83.71** | **71.78** | **58.58** | **32.45** |

During this training epoch, batched instances $\mathcal{D}_t$ are sampled from the reduced dataset $\mathcal{D} \setminus \mathcal{D}'_\rho$, which are then employed for pre-training using contrastive learning (Eqn. 1). This bootstrapping process provides us with robust estimates regarding data samples and enables us to refrain from making strict assumptions about the underlying distribution of the data (Efron, 2003; Sivaganesan, 1994).

In each iterative round, where $\rho_{cur}$ increases from 0 to $2\rho$ (where 0 corresponds to the pruned data preparation epoch), the average pruning ratio remains fixed at $\rho$ as predefined. Finally, we grow back to the original full dataset for another round of pruning data preparation and mutation (Fig. 2).

### 3.4 TIME-EFFICIENCY OF SCAN

In fact, our proposed SCAN method introduces negligible additional pre-training time in comparison to each base model. The potentially increased time involves three major steps: metric selection, pruned data preparation, and pruned data retrieval. Since we directly utilize individual loss values, the metric selection step does not impose an additional time overhead. Second, we obtain the candidate pruned data from in-batch samples, typically containing only a few thousand data points, thereby enabling a fast sorting process. Last, retrieving from millions of pruned data sets also leads to negligible costs, as evidenced by our empirical observations.

More importantly, we maintain consistency in the number of training epochs for our SCAN model, aligning it with the epochs utilized by each respective base model. This approach ensures time efficiency within our proposed method, as demonstrated in Sect. 4.5.

## 4 EXPERIMENTS

### 4.1 EXPERIMENTAL SETTINGS

**Pre-Training Datasets.** For **CLIP** models, we utilized two versions of pre-training datasets to examine the data-size scaling law as well. We employed the OpenCLIP repository (Ilharco et al., 2021) to *conduct pre-training for all models (including CLIP) from scratch*, ensuring a fair comparison between our proposed method and the baselines. Specifically, the smaller dataset, denoted as **CC3M+**,

Table 2: Performance comparison of MoCo models. MoCo utilizes **1.28M** pre-trained data, while the remaining models use **0.83M**. The best results (excluding the original MoCo model) are highlighted in **bold**.

| Arc | Method | ImageNet | | CIFAR-100 | |
|---|---|---|---|---|---|
| | | Top-1 | Top-5 | Top-1 | Top-5 |
| ViT-S/16 | MoCo (Chen et al., 2021) | 78.48 | 94.17 | 86.02 | 97.85 |
| | Random | 75.38 | 91.18 | 84.00 | 95.59 |
| | Info-Batch (Qin et al., 2024) | 77.99 | 93.45 | 85.51 | 97.49 |
| | SCAN | **78.58** | **94.19** | **86.01** | **97.53** |
| ViT-B/16 | MoCo (Chen et al., 2021) | 79.53 | 94.49 | 88.31 | 98.04 |
| | Random | 75.28 | 91.01 | 85.99 | 97.02 |
| | Info-Batch (Qin et al., 2024) | 78.46 | 94.18 | 87.69 | 97.70 |
| | SCAN | **79.15** | **94.33** | **88.11** | **97.78** |

Table 3: Performance comparison of CLIP models on the **CC3M+** pre-trained datasets. CLIP utilizes **4.1M** pre-trained data samples, while the remaining models use **2.9M**. The best results (excluding the original CLIP model) are highlighted in **bold**. A dash (-) indicates the collapse of pre-training, resulting in impaired evaluation of downstream tasks.

| Architecture | Method | IN Zero-Shot | | CIFAR10 | CIFAR100 | IN | IN-V2 | IN-R |
|---|---|---|---|---|---|---|---|---|
| | | Top-1 | Top-5 | | | | | |
| RN101 | CLIP | 15.72 | 35.19 | 96.17 | 81.78 | 75.39 | 63.42 | 39.69 |
| | Random | 12.35 | 29.03 | 95.01 | 78.99 | 73.73 | 61.01 | 36.11 |
| | SemDeDup | 12.97 | 28.90 | 95.16 | 79.44 | 74.08 | 61.94 | 36.74 |
| | D-Pruning | 12.77 | 28.03 | 94.85 | 78.23 | 73.55 | 61.56 | 36.48 |
| | Info-Batch | **16.79** | 34.38 | 95.49 | 80.89 | 74.48 | 62.11 | 38.01 |
| | SCAN | 15.59 | **34.77** | **95.77** | **81.95** | **74.61** | **62.92** | **38.05** |
| ViT-B/16 | CLIP | 18.17 | 37.62 | 96.58 | 82.47 | 70.78 | 57.28 | 30.13 |
| | Random | 13.26 | 31.27 | 91.62 | 73.53 | 50.60 | 40.55 | 21.80 |
| | SemDeDup | 11.35 | 25.56 | 94.36 | 76.53 | 66.56 | 53.18 | 25.50 |
| | D-Pruning | 10.00 | 23.31 | 93.46 | 75.37 | 65.80 | 52.50 | 24.39 |
| | Info-Batch | 17.16 | **39.14** | 95.98 | 81.43 | 69.19 | 56.00 | 28.55 |
| | SCAN | **18.21** | 37.20 | **96.00** | **81.49** | **69.46** | **56.04** | **28.60** |
| Swin-Base | CLIP | 13.98 | 32.05 | 95.75 | 82.19 | 73.89 | 61.92 | 35.38 |
| | Random | 12.56 | 30.12 | 94.10 | 78.01 | 72.55 | 60.25 | 31.31 |
| | SemDeDup | - | - | - | - | - | - | - |
| | D-Pruning | 12.44 | 28.60 | 94.07 | 77.86 | 72.54 | 60.28 | 31.41 |
| | Info-Batch | 13.82 | 32.05 | **95.80** | 81.25 | 72.69 | 59.47 | 33.13 |
| | SCAN | **17.50** | **37.42** | 95.37 | **81.35** | **73.55** | **61.60** | **33.55** |

comprises CC3M (Sharma et al., 2018), SBU-Captions (Ordonez et al., 2011), and MSCOCO (Lin et al., 2014), totaling 4.1 million image-text pairs. The larger dataset, denoted as **CC12M+**, includes CC12M (Changpinyo et al., 2021), SBU-Captions (Ordonez et al., 2011), and MSCOCO (Lin et al., 2014), with a total of 10.1 million pairs.

For the pre-training dataset of **MoCo** (Chen et al., 2020c; 2021), we adhered to the original implementation and utilized the ImageNet dataset (Deng et al., 2009).

**Downstream Datasets.** We conducted extensive downstream fine-tuning experiments across various datasets. Specifically, we utilized datasets such as ImageNet (Deng et al., 2009), CIFAR-10, CIFAR-100 (Krizhevsky et al., 2009), as well as out-of-distribution datasets including ImageNet V2 (Recht et al., 2019) and ImageNet-R (Hendrycks et al., 2021) to validate the downstream performance of **CLIP** pre-trained models. For all these datasets, we explored diverse experimental settings, encompassing zero-shot transfer learning from ImageNet, linear probing, and full fine-tuning. Moreover, we employed both the ImageNet and CIFAR-100 datasets to conduct experiments on **MoCo** models.

**Model Architectures.** As OpenCLIP (Ilharco et al., 2021) offers a variety of model cards, we utilized model architectures from both ResNet (He et al., 2016) and ViT (Dosovitskiy et al., 2021). The model architectures employed for pre-training **CLIP** include RN50, RN101, ViT-S/32, ViT-S/16, ViT-B/32, ViT-B/16, and Swin-Base (Liu et al., 2021b). Due to resource constraints, we pre-trained **MoCo** using two model architectures: ViT-S/16 and ViT-B/16 (Touvron et al., 2021). *It is important to note that pre-training MoCo consumes approximately seven times more resources than pre-training a CLIP model. Therefore, we primarily conducted experiments on CLIP to assess the effectiveness of the proposed method.*

**Compared Baselines.** Given the absence of common models for addressing the data efficiency problem in contrastive pre-training, we adapted four different approaches in this study: Random, SemDeDup (Abbas et al., 2023), D-Pruning (Yang et al., 2023), and Info-Batch (Qin et al., 2024). Unless otherwise specified, we utilized a pruning ratio of 30% for CLIP models and 35% for MoCo models. Additionally, we provide more details regarding the baseline introduction and training protocol in Appendix B.2 and B.1.

## 4.2 OVERALL EXPERIMENTAL RESULTS

We present the comprehensive results of CLIP in Tables 1 and 3, and the results of MoCo in Table 2. Additional experimental results can be found in the appendix. The insights from these tables can be summarized into four key observations:

- Our proposed SCAN method consistently outperforms the four compared baselines, indicating that our approach achieves a superior trade-off between performance and data efficiency compared to existing data-efficient methods.

- In comparison to the base CLIP models, SCAN achieves comparable performance while utilizing fewer data for pre-training. Specifically, our method preserves 99% of the original CLIP model's performance in most cases, while using only 30% of the original training dataset. Take the IN result of the RN101 architecture in Table 3 as an example: 74.61 (SACN) *v.s.* 75.39 (CLIP) - 99% of CLIP result. It is also worth noting that some of our methods even outperform the base CLIP models.

- Dynamic approaches such as Info-Batch and SCAN often outperform static coreset selection methods like SemDeDup and D-Pruning. This verifies the superiority of dynamic pruning approaches.

- An apple-to-apple comparison between Table 1 and Table 3 reveals that models trained with more data (CC12M+) consistently outperform those trained with less data (CC3M+), thereby verifying the dataset scaling law (Hoffmann et al., 2022; Kaplan et al., 2020).

## 4.3 CORESET RESULTS FROM SCAN

Beyond the dynamic data pruning approach, we also investigated the coreset results obtained from our SCAN method. To implement this, we first identify the pruned data using two pre-trained models from our method, such as RN50 and ViT-S/16. After that, we obtain the intersection of these two sets and ensure that the overall pruning ratio remains the same as $\rho$, thereby generating a coreset from the original full dataset.

**Downstream Performance.** We then performed pre-training for another model from scratch, *e.g.*, ViT-B/32, employing the exact same process as previous static coreset-based approaches (Abbas et al.,

Table 4: Coreset selection model results on the ImageNet dataset. The coreset generated by our SCAN (static) method is derived from the intersection of the datasets obtained from the other two models. All models are pre-trained from scratch using each respective coreset.

| Method | RN50 | | | ViT-S/16 | | | ViT-B/32 | | |
|---|---|---|---|---|---|---|---|---|---|
| | ZS@1 | ZS@5 | FT@1 | ZS@1 | ZS@5 | FT@1 | ZS@1 | ZS@5 | FT@1 |
| SCAN | 16.91 | 35.79 | 72.91 | 17.31 | 35.51 | 66.86 | 16.48 | 33.60 | 56.64 |
| SemDeDup | 11.98 | 26.30 | 71.51 | 09.57 | 22.00 | 62.30 | 07.20 | 17.50 | 50.99 |
| D-Pruning | 11.72 | 26.65 | 71.11 | 08.60 | 20.35 | 61.70 | 06.51 | 16.13 | 50.01 |
| SCAN (static) | 16.99 | 35.20 | 73.11 | 16.68 | 34.31 | 66.22 | 13.16 | 28.46 | 55.99 |

2023; Yang et al., 2023). From the results in Table 4, we observe the following: I) Our selected coreset significantly outperforms other existing coreset-based baselines utilizing using the exact same settings. II) Our static method achieves very competitive results compared to our dynamic variant. This observation further underscores the potential of our proposed method for identifying a subset that can be directly and efficiently utilized in future research endeavors, thereby reducing more training overhead.

**Coreset Overlaps.** We then calculated the intersection over union for each pair and trio of coresets. The results are presented in Table 5. The result reveals that I) The coresets obtained are highly related to the specific model architecture used. II) The coresets from ViT-B/32 and ViT-S/16 have a higher degree of overlap than the other two, as these two models share the same architecture family.

Table 5: Coreset overlap ratios.

| Model | | | Overlap |
| RN50 | ViT-S/16 | ViT-B/32 | Ratio |
|---|---|---|---|
| ✓ | ✓ | | 56.78% |
| ✓ | | ✓ | 55.77% |
| | ✓ | ✓ | 61.73% |
| ✓ | ✓ | ✓ | 47.89% |

## 4.4 ABLATION STUDY

**Different Pruning Ratios.** The performance change corresponding to different pruning ratios $\rho$ is depicted in Fig. 3. Generally, we can observe that the performance tends to degrade with increasing pruning ratios. Further, selecting the optimal pruning ratio also involves a trade-off.

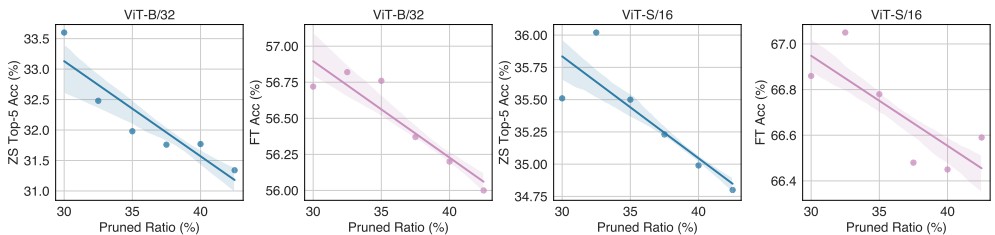

Figure 3: Downstream performance variation of two CLIP models *w.r.t.* different pruning ratios.

**Different Mutation Epochs.** The results for different mutation epochs are summarized in Table 6. It can be observed that employing a mutation epoch of three generally yields better results compared to other variants.

Table 6: Results *w.r.t.* mutation epoch.

| Mutation | ViT-S/32 | | ViT-S/16 | |
| Epochs | ZS@5 | FT@1 | ZS@5 | FT@1 |
|---|---|---|---|---|
| 2 | 28.72 | 56.38 | **36.28** | 66.81 |
| 3 | **33.60** | **56.72** | 35.51 | **67.04** |
| 4 | 28.31 | 56.25 | 33.80 | 66.94 |
| 5 | 29.43 | **58.33** | 33.29 | 67.02 |

**Different Pruning Modes.** Our SCAN method combines samples categorized as both *redundant* and *ill-matched* for pruning purposes. Additionally, we conducted a separate analysis to evaluate the effectiveness of utilizing either *redundant* (**R.**) or *ill-matched* (**I.**) samples alone, and the results are presented in Table 7. It is evident from the table that employing *ill-matched* samples for pruning yields superior performance compared to using *redundant* samples alone. Moreover, the combination of these two categories results in further performance improvement.

Table 7: Results *w.r.t.* different pruning modes.

| Pruning Mode | | ViT-S/32 | | ViT-S/16 | |
| *w/.* R. | *w/.* I. | ZS@5 | FT@1 | ZS@5 | FT@1 |
|---|---|---|---|---|---|
| ✓ | ✗ | 31.48 | 56.66 | 33.51 | 66.85 |
| ✗ | ✓ | **34.12** | 56.01 | 35.19 | 67.04 |
| ✓ | ✓ | 33.60 | **56.72** | **35.51** | **67.04** |

Table 8: Results *w.r.t.* distinct ratio variants.

| R. *v.s.* I. (%) | ViT-S/32 | | ViT-S/16 | |
| | ZS@5 | FT@1 | ZS@5 | FT@1 |
|---|---|---|---|---|
| (30 : 10) | 31.84 | 56.35 | 34.84 | 66.90 |
| (20 : 20) | 33.60 | **56.72** | **35.51** | 67.04 |
| (10 : 30) | **33.72** | 56.43 | 35.49 | **67.10** |

**Different Variants of the Same Pruning Ratio.** In our experiments, we employed a pruning ratio $\rho$ of 30% for CLIP models and divided it equally between redundant (**R.**) and ill-matched (**I.**) samples. Furthermore, we investigated other variants, and the outcomes are presented in Table 8. All the ratios are relative to the full dataset. It can be seen from the table that evenly distributing the pruning ratio leads to slightly better results.

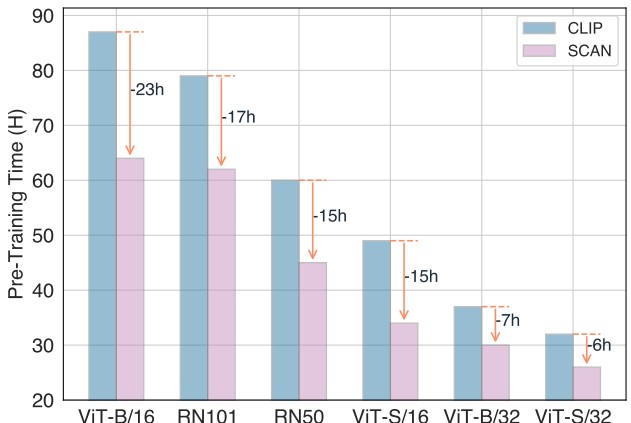

Figure 4: Comparison of pre-training time between the base CLIP model and our SCAN on the CC12M+ dataset.

Figure 5: Ill-matched examples.

### 4.5 In-depth Analysis

**Pre-Training Time Comparison.** The primary outcome of our method is the reduction in training time and, consequently, the decrease in carbon footprint. We illustrate the pre-training time in Fig. 4. It can be observed from the figure that our SCAN method contributes to a reduction of approximately 25% to 30% in the original pre-training computational cost (The additional negligible time cost can be attributed to the pruned data preparation and retrieval processes.). This advantage proves especially beneficial when training large models such as RN101 and ViT-B-16.

**Visualization of Ill-Matched Samples.** We also provide visualizations of some ill-matched samples identified by SCAN. Two examples are shown in Fig. 5. In the first example, an incorrect annotation is evident, as there is no *city* depicted in the image. Additionally, in the second example, the individual portrayed is identified as a *doctor* rather than a *stockman*.

## 5 Conclusion

**Limitations.** We acknowledge two potential limitations of this work. Firstly, akin to other dataset pruning methodologies, our method necessitates the storage of the original large-scale dataset. This may pose storage challenges for researchers with limited computing resources. Secondly, our method may not seamlessly transfer to the recent popular large language model pre-training. Apart from differences in pre-training objectives, large language models (LLMs) often require only a few training epochs, typically one to three. In contrast, our iterative bootstrapping strategy requires more epochs to converge, the same as each corresponding contrastive pre-training model.

**Summary.** This work sets an initial effort to comprehensively address the data efficiency challenge in contrastive pre-training. We propose a novel dataset bootstrapping approach, applying it to a range of contrastive pre-training model architectures and evaluating it using various protocols. Our experiments demonstrate that, across all experimental settings, the proposed method achieves a superior balance between downstream model performance and data efficiency compared to both the base models and several existing data efficiency approaches. Additionally, it also helps yield an effective coreset dataset that significantly outperforms other coreset-based baselines, thereby further reducing time costs and training overhead.

**Future Work.** In the future, we plan to validate the effectiveness of our method 1) in more domains such as language-centric contrastive pre-training, 2) with larger pre-training datasets for vision-language like LAION-400M (Schuhmann et al., 2021). Moreover, our method stands orthogonal to other efficiency studies, such as model compression. As such, by integrating strategies from these related domains, we aim to build a more efficient training pipeline framework, thus contributing to substantial reductions in carbon footprints.

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

## A  SCAN ALGORITHM

We present a detailed algorithm of our proposed SCAN in Algorithm A. This algorithm is applicable to contrastive pre-training models including CLIP and MoCo.

---

**Algorithm 1:** Dataset Pruning of SCAN.

---

**Input:** Full training data $\mathcal{D}$, Number of training epochs $\tau_{stop}$, Number of mutation epochs $\tau_{cos}$,
   Pre-initialized losses $\mathcal{L}_{pre}$ and $\mathcal{L}_{cur}$, Threshold value $T_{td}$ and an infinitesimal value $\epsilon$.
**Output:** Pre-trained model $\mathcal{M}$
**while** $\tau_{cur} < \tau_{stop}$ **do**
    `// Pre-Pruning Warm-Up`
    **if** $(\hat{\mathcal{L}}_{pre} - \hat{\mathcal{L}}_{cur})/(\hat{\mathcal{L}}_{pre} + \epsilon) \geq T_{td}$ **then**
        **for** *Batched sample* $\mathcal{D}_t \in \mathcal{D}$ **do**
            | Forward and update $\mathcal{M}$ on $\mathcal{D}_t$;
        **end**
        $\hat{\mathcal{L}}_{pre} \leftarrow \hat{\mathcal{L}}_{cur}$;
        Get the updated current epoch loss $\hat{\mathcal{L}}_{cur}$;
    **end**
    **else**
        `// Pruning Data Preparation`
        **if** $\tau_{cur}\ mod\ (\tau_{cos} + 1) = 0$ **then**
            **for** *Batched sample* $\mathcal{D}_t \in \mathcal{D}$ **do**
                Forward and update $\mathcal{M}$ on $\mathcal{D}_t$;
                Obtain *redundant* set $\mathcal{D}_t^{red}$ and *ill-matched* set $\mathcal{D}_t^{ill}$;
                Obtain the overall pruning subset $\mathcal{D}_t^{'} = \mathcal{D}_t^{red} | \mathcal{D}_t^{ill}$;
            **end**
            Accumulate all the candidate pruning data $\mathcal{D}^{'}$;
        **end**
        `// Dataset Mutation`
        **else**
            Obtain the pruning ratio $\rho_{cur}$;
            Randomly prune $\rho_{cur}|\mathcal{D}^{'}|$ samples from $\mathcal{D}^{'}$;
            **for** *Batched sample* $\mathcal{D}_t \in \mathcal{D} \setminus \mathcal{D}_{\rho}^{'}$ **do**
                | Forward and update $\mathcal{M}$ on $\mathcal{D}_t$
            **end**
        **end**
    **end**
    $\tau_{cur} \leftarrow \tau_{cur} + 1$
**end**

---

## B  MORE EXPERIMENTAL SETTINGS

### B.1  PRE-TRAINING DETAILS

Our primary objective in this study is to assess the efficacy of our proposed data-efficient method. Consequently, we did not conduct an extensive parameter search and instead utilized a universal setting across different models.

Table 9: Batch sizes for pre-training and fine-tuning CLIP models.

| PT | RN50 | RN101 | ViT-S/32 | ViT-S/16 | ViT-B/32 | ViT-B/16 | Swin-Base |
|----|------|-------|----------|----------|----------|----------|-----------|
| ✓  | 256×4 | 200×4 | 800×4 | 400×4 | 480×4 | 200×4 | 100×4 |
| ✗  | 384 | 225 | 1024 | 600 | 768 | 300 | 160 |

Due to limitations in computational resources, most of our pre-training experiments were conducted using four NVIDIA A5000 GPUs. Specifically, for CLIP models, we employed 32 epochs, a learning rate of 1e-3, and a weight decay of 0.1. Various batch sizes are detailed in Table 9. For the downstream image classification task, we fine-tuned the pre-trained models on a single NVIDIA A100-40G GPU. Fine-tuning comprises 10 epochs with a learning rate of 1e-3 and a weight decay of 0.1.

Regarding the pre-training of MoCo, we utilized the original implementation[3]. We employed batch sizes of 600 and 370 for ViT-16/S and ViT-B/16, respectively.

## B.2 COMPARED BASELINES

We compared with the following four baselines in this work:

- **Random** prunes $\rho$ samples with randomness for each epoch. Notably, it falls under dynamic pruning methods as the pruned samples vary across epochs.
- **SemDeDup** (Abbas et al., 2023) identifies the semantic duplicates based on embedding similarities. We used one public implementation[4]. This method is applicable only to multi-modal models such as CLIP.
- **D-Pruning** (Yang et al., 2023) estimate the parameter influence of a training example through the removal of it. We utilized the official implementation[5] for CLIP models only. We abandoned the use of MoCo due to its hard-to-configure running environment.
- **Info-Batch** (Qin et al., 2024) is a recent robust dataset pruning baseline. It prunes a portion of less informative samples and then rescales the gradients of the remaining samples to approximate the original gradients. We followed the original code[6] to re-implement it for our experiments.

## C MORE EXPERIMENTAL RESULTS

We present additional fine-tuning results of CLIP in Table 10 and Table 11. Furthermore, Table 12 shows the results of linear probing for CLIP. It is evident that our proposed SCAN method consistently achieves superior performance across various settings.

**Experimental Results on CLIP-Benchmark.** We utilized the CLIP-Benchmark tool to assess the performance of both CLIP and our SCAN method across 19 additional datasets. For this evaluation, we employed models pre-trained on the CC12M+ datasets. The results, presented in Table 13, demonstrate that our SCAN method delivers performance competitive with the original CLIP.

**Results *w.r.t.* Pre-defined Thresholds.** To assess the impact of varying thresholds, we evaluated two model architectures, RN50 and ViT-B/32, using threshold values from 0.1 to 0.7, with a step size of 0.2. The ImageNet zero-shot performance results are summarized in the table below. As indicated, the models perform optimally at threshold values of 0.3 or 0.5. For simplicity and consistency, we selected a threshold of 0.3 for subsequent model evaluations.

**Different Pruning Ratios of MoCo.** The performance variations with different pruning ratios ($\rho$) for the MoCo model are depicted in Fig. 6. It is evident that as the pruning ratios increase, there is a general degradation in performance.

**More Visualization of Ill-matched Samples from CLIP.** We further visualize some ill-matched samples as indicated by SCAN in Fig. 7.

---

[3]https://github.com/facebookresearch/moco-v3.
[4]https://github.com/BAAI-DCAI/Dataset-Pruning/tree/main.
[5]https://github.com/BAAI-DCAI/Dataset-Pruning/tree/main.
[6]https://github.com/henryqin1997/InfoBatch.

Table 10: Performance comparison of CLIP models on the **CC3M+** pre-trained datasets. CLIP utilizes **4.1M** pre-trained data samples, while the remaining models use **2.9M**. The best results (excluding the original CLIP model) are highlighted in **bold**.

| Architecture | Method | IN Zero-Shot | | CIFAR10 | CIFAR100 | IN | IN-V2 | IN-R |
| --- | --- | --- | --- | --- | --- | --- | --- | --- |
| | | Top-1 | Top-5 | | | | | |
| RN50 | CLIP | 17.06 | 36.21 | 95.32 | 80.01 | 73.81 | 61.89 | 36.09 |
| | Random | 11.02 | 25.23 | 94.01 | 75.12 | 70.22 | 58.04 | 31.80 |
| | SemDeDup | 11.98 | 26.30 | 94.53 | 76.81 | 71.51 | 58.79 | 32.31 |
| | D-Pruning | 11.72 | 26.65 | 94.48 | 76.73 | 71.11 | 58.79 | 31.88 |
| | Info-Batch | 16.44 | **36.74** | 95.30 | 79.40 | **73.01** | **61.49** | **35.04** |
| | SCAN | **16.91** | 35.79 | **95.30** | **80.24** | 72.91 | 60.59 | 34.53 |
| ViT-S/32 | CLIP | 13.70 | 29.33 | 90.59 | 71.74 | 55.60 | 42.81 | 23.91 |
| | Random | 06.57 | 16.19 | 86.61 | 60.18 | 48.87 | 34.48 | 17.98 |
| | SemDeDup | 05.33 | 14.05 | 85.16 | 59.87 | 47.39 | 35.56 | 17.70 |
| | D-Pruning | 04.78 | 12.91 | 84.21 | 57.96 | 46.53 | 34.77 | 16.88 |
| | Info-Batch | 10.89 | 26.91 | 90.02 | 69.99 | 50.53 | 39.61 | 19.69 |
| | SCAN | **14.88** | **31.47** | **90.12** | **70.33** | **54.13** | **41.29** | **22.70** |
| ViT-S/16 | CLIP | 18.41 | 37.41 | 96.09 | 81.31 | 68.49 | 55.79 | 29.52 |
| | Random | 07.80 | 21.53 | 93.58 | 72.11 | 62.13 | 49.63 | 19.01 |
| | SemDeDup | 09.57 | 22.00 | 93.43 | 74.37 | 62.30 | 48.89 | 23.04 |
| | D-Pruning | 08.60 | 20.35 | 93.26 | 73.72 | 61.70 | 48.97 | 22.46 |
| | Info-Batch | 16.19 | 35.06 | **95.64** | 80.03 | 67.57 | 53.52 | **27.64** |
| | SCAN | **17.31** | **35.51** | 95.53 | **80.27** | **66.86** | **53.59** | 27.34 |
| ViT-B/32 | CLIP | 14.97 | 32.02 | 94.43 | 77.72 | 58.33 | 45.70 | 25.59 |
| | Random | 07.44 | 18.88 | 89.96 | 69.41 | 50.43 | 40.62 | 18.07 |
| | SemDeDup | 07.20 | 17.50 | 90.88 | 70.13 | 50.99 | 38.34 | 19.76 |
| | D-Pruning | 06.51 | 16.13 | 60.07 | 69.11 | 50.01 | 38.43 | 19.03 |
| | Info-Batch | 12.44 | 30.98 | 93.57 | 75.44 | 55.99 | 43.30 | **24.64** |
| | SCAN | **16.48** | **33.60** | **93.77** | **77.63** | **56.64** | **44.25** | 24.10 |

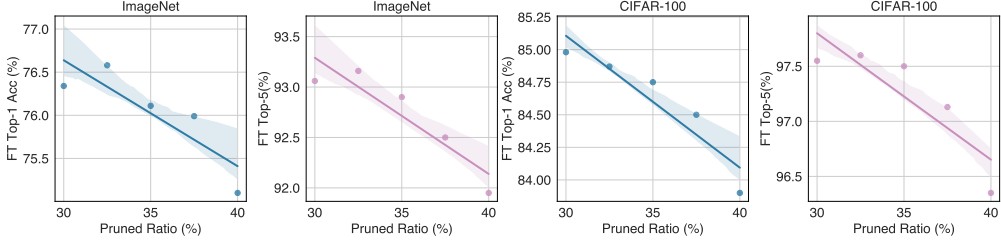

Figure 6: Downstream performance variation of ViT-S/16 MoCo model *w.r.t.* different pruning ratios.

Table 11: Performance comparison of CLIP models on the **CC12M+** pre-trained datasets. CLIP utilizes **10.1M** pre-trained data samples, while the remaining models use **7.1M**. The best results (excluding the original CLIP model) are highlighted in **bold**.

| Architecture | Method | IN Zero-Shot | | CIFAR10 | CIFAR100 | IN | IN-V2 | IN-R |
|---|---|---|---|---|---|---|---|---|
| | | Top-1 | Top-5 | | | | | |
| RN50 | CLIP | 20.95 | 44.41 | 95.68 | 80.75 | 74.93 | 62.81 | 38.36 |
| | Random | 12.39 | 35.96 | 94.89 | 76.96 | 71.65 | 59.71 | 32.03 |
| | SemDeDup | 15.89 | 36.76 | 95.00 | 78.12 | 72.46 | 60.01 | 33.86 |
| | D-Pruning | 11.19 | 26.53 | 94.31 | 77.69 | 71.96 | 59.19 | 33.44 |
| | Info-Batch | 20.63 | 45.10 | **95.68** | 79.88 | 73.53 | 61.23 | 36.67 |
| | SCAN | 23.03 | 47.83 | 95.63 | **81.03** | 74.28 | 62.20 | 38.14 |
| ViT-S/32 | CLIP | 26.48 | 51.32 | 93.23 | 76.32 | 61.53 | 48.60 | 30.57 |
| | Random | 08.79 | 16.93 | 87.79 | 63.04 | 50.12 | 38.09 | 21.11 |
| | SemDeDup | 05.04 | 13.49 | 86.43 | 61.67 | 49.46 | 37.37 | 19.29 |
| | D-Pruning | 04.54 | 12.43 | 85.86 | 61.81 | 48.39 | 36.57 | 18.62 |
| | Info-Batch | 10.07 | 26.63 | 91.11 | 67.94 | 53.47 | 40.91 | 20.77 |
| | SCAN | 25.27 | 50.08 | 91.86 | 75.27 | 59.87 | 46.96 | 27.86 |
| ViT-S/16 | CLIP | 27.09 | 53.57 | 96.62 | 84.05 | 71.40 | 58.40 | 34.24 |
| | Random | 16.58 | 35.43 | 95.00 | 79.90 | 67.78 | 54.12 | 26.23 |
| | SemDeDup | 10.56 | 26.52 | 94.46 | 76.65 | 65.32 | 51.37 | 25.52 |
| | D-Pruning | 09.37 | 22.16 | 93.42 | 75.52 | 63.53 | 50.79 | 24.43 |
| | Info-Batch | 21.28 | 45.56 | 96.09 | 82.13 | 68.87 | 55.90 | 29.58 |
| | SCAN | 28.46 | 54.56 | 96.24 | 83.32 | 70.40 | 57.10 | 31.85 |

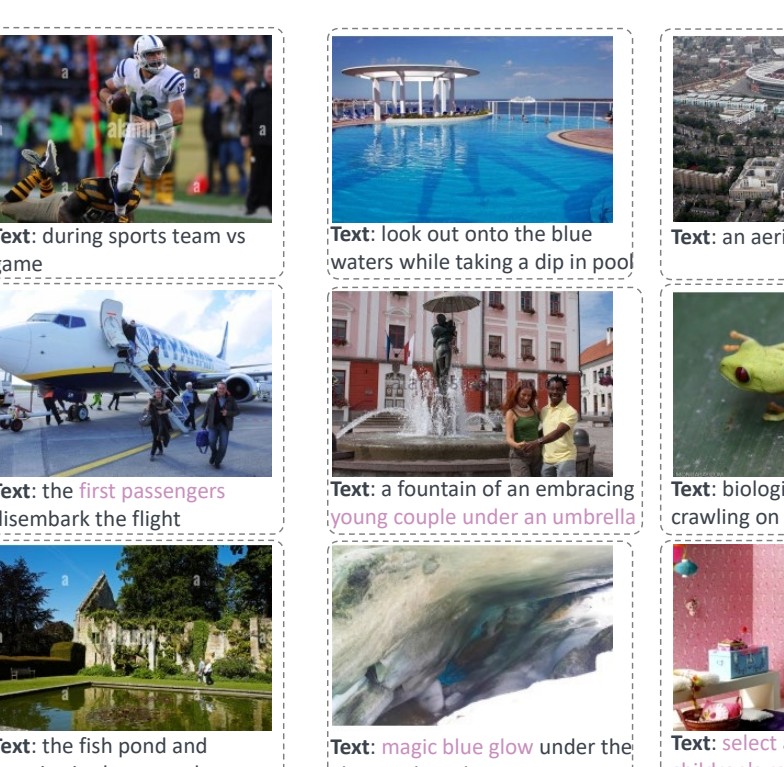

Figure 7: More *ill-matched* samples obtained by our SCAN approach.

Table 12: Linear probing results of six CLIP models. For the CC3M+ pre-trained datasets, CLIP utilizes **4.1M** pre-trained data samples, while the remaining models use **2.9M**. For the CC12M+ pre-trained datasets, CLIP utilizes **10.1M** pre-trained data samples, while the remaining models use **7.1M**. The best results (excluding the original CLIP model) are highlighted in **bold**. A dash (-) indicates the collapse of pre-training, resulting in impaired evaluation of downstream tasks.

| Arc | Method | CC3M+ | | | | | CC12M+ | | | | |
|---|---|---|---|---|---|---|---|---|---|---|---|
| | | CF-10 | CF-100 | IN | IN-V2 | IN-R | CF-10 | CF-100 | IN | IN-V2 | IN-R |
| RN50 | CLIP | 95.58 | 80.31 | 73.96 | 61.60 | 35.59 | 95.69 | 81.88 | 74.96 | 62.85 | 38.57 |
| | Random | 93.89 | 75.45 | 70.25 | 58.05 | 31.78 | 94.00 | 76.43 | 70.99 | 58.78 | 32.09 |
| | SemDeDup | 94.92 | 77.16 | 71.62 | 58.99 | 32.44 | 94.88 | 78.00 | 72.22 | 59.70 | 33.16 |
| | D-Pruning | 94.50 | 76.78 | 71.00 | 57.98 | 31.70 | 94.30 | 77.70 | 71.77 | 59.01 | 33.20 |
| | Info-Batch | 95.29 | 79.39 | 73.07 | 61.03 | **34.66** | **95.66** | 79.84 | 73.23 | 61.10 | 36.63 |
| | SCAN | **95.46** | **80.35** | **73.07** | **61.25** | 34.59 | 95.62 | **81.28** | **74.27** | **62.66** | **37.30** |
| RN101 | CLIP | 95.92 | 82.04 | 75.10 | 63.61 | 38.78 | 96.03 | 82.73 | 75.78 | 63.93 | 40.09 |
| | Random | 95.00 | 78.13 | 73.79 | 60.20 | 36.12 | 95.02 | 78.34 | 73.99 | 60.27 | 36.13 |
| | SemDeDup | 94.84 | 79.25 | 74.08 | 61.94 | 36.74 | 95.01 | 78.02 | 73.89 | 59.91 | 33.80 |
| | D-Pruning | 94.79 | 72.12 | 73.74 | 61.66 | 35.64 | 94.78 | 78.83 | 74.08 | 61.28 | 37.09 |
| | Info-Batch | 95.08 | 80.76 | 74.13 | 62.89 | 37.57 | 95.82 | 81.56 | 75.02 | 63.21 | 39.21 |
| | SCAN | **95.67** | **81.36** | **74.42** | **63.07** | **37.86** | **95.93** | **82.12** | **75.61** | **63.87** | **39.32** |
| ViT-S/32 | CLIP | 91.65 | 72.23 | 55.52 | 43.00 | 23.48 | 93.29 | 77.06 | 61.73 | 48.84 | 30.40 |
| | Random | 87.00 | 61.31 | 49.97 | 36.07 | 20.88 | 87.79 | 63.04 | 50.12 | 38.09 | 21.11 |
| | SemDeDup | 83.46 | 60.06 | 47.65 | 35.51 | 17.61 | 86.23 | 61.77 | 49.20 | 37.10 | 19.11 |
| | D-Pruning | 84.21 | 58.73 | 46.57 | 35.03 | 16.95 | 85.82 | 61.09 | 47.99 | 36.58 | 18.00 |
| | Info-Batch | 89.30 | 70.02 | 50.51 | 39.58 | 19.78 | 91.02 | 68.90 | 53.49 | 40.69 | 20.71 |
| | SCAN | **89.37** | **71.05** | **54.24** | **41.30** | **22.65** | **91.88** | **74.86** | **59.90** | **46.90** | **27.90** |
| ViT-S/16 | CLIP | 96.09 | 81.39 | 68.49 | 55.19 | 29.06 | 96.66 | 84.35 | 71.53 | 58.56 | 33.85 |
| | Random | 93.62 | 73.37 | 63.02 | 49.96 | 20.62 | 94.90 | 79.91 | 67.90 | 54.10 | 26.24 |
| | SemDeDup | 93.21 | 73.85 | 62.34 | 49.40 | 22.54 | 94.00 | 77.01 | 64.45 | 51.40 | 25.51 |
| | D-Pruning | 93.28 | 73.09 | 61.67 | 48.99 | 22.48 | 93.41 | 75.43 | 63.42 | 50.77 | 24.41 |
| | Info-Batch | 95.26 | **80.46** | **67.76** | 53.49 | 27.11 | 96.03 | 82.11 | 68.78 | 55.78 | 29.59 |
| | SCAN | **95.31** | 80.00 | 67.04 | **53.75** | **27.41** | **96.37** | **82.71** | **70.32** | **57.17** | **31.89** |
| ViT-B/32 | CLIP | 94.36 | 77.84 | 58.43 | 45.79 | 25.50 | 95.65 | 81.62 | 63.40 | 50.33 | 31.28 |
| | Random | 90.05 | 69.26 | 50.23 | 40.54 | 18.03 | 90.13 | 69.98 | 51.99 | 41.01 | 20.08 |
| | SemDeDup | 90.44 | 69.86 | 50.89 | 38.15 | 19.89 | 90.77 | 70.00 | 51.19 | 39.80 | 20.91 |
| | D-Pruning | 90.06 | 69.08 | 50.04 | 37.87 | 19.11 | 90.07 | 69.65 | 51.23 | 37.99 | 20.43 |
| | Info-Batch | 93.54 | 75.49 | **56.98** | 44.03 | 24.08 | - | - | - | - | - |
| | SCAN | **94.00** | **76.91** | 56.72 | **44.12** | **24.21** | **95.05** | **81.21** | **61.96** | **48.42** | **29.53** |
| ViT-B/16 | CLIP | 96.27 | 82.74 | 70.87 | 57.77 | 29.82 | 96.77 | 84.48 | 72.37 | 59.07 | 33.24 |
| | Random | 91.60 | 73.61 | 50.59 | 40.52 | 21.72 | 94.56 | 76.67 | 67.57 | 54.40 | 27.10 |
| | SemDeDup | 94.16 | 76.34 | 66.60 | 53.13 | 25.60 | 94.17 | 76.66 | 67.10 | 53.39 | 27.11 |
| | D-Pruning | 93.48 | 75.41 | 65.90 | 52.69 | 24.57 | 93.88 | 75.99 | 65.98 | 53.00 | 26.05 |
| | Info-Batch | 96.10 | 81.06 | **70.30** | 56.10 | 28.48 | 96.12 | 81.78 | 71.34 | 56.25 | 31.12 |
| | SCAN | **96.16** | **81.10** | 69.55 | **56.48** | **28.76** | **96.12** | **83.97** | **71.82** | **58.31** | **32.48** |

Table 13: Comparison of ViT-B/32 and ViT-B/16 using CLIP and SCAN on CLIP-Benchmark.

| Dataset | ViT-B/32 | | ViT-B/16 | |
| --- | --- | --- | --- | --- |
| | CLIP | SCAN | CLIP | SCAN |
| FER2013 | 18.50 | 22.27 | 18.36 | 20.77 |
| ImageNet-O | 30.70 | 30.55 | 33.05 | 31.20 |
| ImageNet-R | 29.23 | 31.91 | 31.08 | 29.67 |
| ImageNetv2 | 20.19 | 21.80 | 21.39 | 20.90 |
| ObjectNet | 15.13 | 13.93 | 14.84 | 15.03 |
| rendered-sst2 | 50.08 | 49.92 | 51.12 | 50.02 |
| STL-10 | 85.18 | 86.06 | 85.11 | 85.04 |
| SUN397 | 40.55 | 41.02 | 41.95 | 41.29 |
| VOC-2007 | 47.22 | 42.62 | 52.59 | 48.48 |
| Caltech-101 | 64.93 | 68.56 | 65.63 | 65.46 |
| Dmlab | 20.02 | 11.81 | 17.77 | 16.19 |
| DTD | 15.66 | 16.44 | 16.24 | 13.83 |
| EuroSat | 21.92 | 29.81 | 34.20 | 29.67 |
| Flowers | 18.63 | 24.70 | 20.80 | 20.13 |
| KITTI | 32.63 | 32.77 | 35.49 | 35.59 |
| PCam | 50.33 | 52.23 | 50.32 | 52.69 |
| Pet | 31.28 | 43.06 | 36.41 | 35.84 |
| RESISC45 | 23.41 | 23.05 | 21.28 | 19.38 |
| SVHN | 16.99 | 06.97 | 09.73 | 07.86 |

Table 14: Performance comparison of RN50 and ViT-B/32 at different thresholds.

| Threshold | RN50 | | ViT-B/32 | |
| --- | --- | --- | --- | --- |
| | Top-1 | Top-5 | Top-1 | Top-5 |
| 0.1 | 15.80 | 35.21 | 14.75 | 31.58 |
| 0.3 | 16.91 | 35.79 | 16.48 | 33.60 |
| 0.5 | 18.22 | 37.79 | 16.04 | 33.19 |
| 0.7 | 18.20 | 37.78 | 16.48 | 33.23 |

