# OpenReview forum: "SCAN: Bootstrapping Contrastive Pre-training for Data Efficiency"
_ICLR.cc/2025/Conference — ICLR 2025 Conference Withdrawn Submission_

### Official Review · Reviewer_eEQF · 2024-10-26

**Soundness:** 2
**Presentation:** 2
**Contribution:** 2
**Rating:** 3
**Confidence:** 4

**Summary:**

This paper presents a dataset pruning method for contrastive pre-training methods. It designs a dynamic bootstrapping pruning method to dynamic prune the data based on current training losses for each sample. It removes the samples with both largest values and smallest values, since large values may stand for low image-text similarity and small values means the model already fit well on that sample. The training is first conducted on the whole dataset and then use a pruning ratio following cosine annealing strategy. CLIP model is experimented with a combination of CC3M, CC12M, SBU, and MSCOCO, and evaluated on ImageNet zero-shot classification, linear probing, and finetuning. MOCO is experimented with ImageNet dataset for training.

**Strengths:**

-	The idea of using loss values to dynamic prune dataset is interesting.
-	The proposed method shows some performance gain compared to some existing core-set selection methods (Info-Batch, D-Pruning)

**Weaknesses:**

-	The data scale for experiments is relatively small. The results with CLIP only achieve ~26% on ImageNet zero-shot classification, which is far from the original work with 70+% on a much larger dataset. The effectiveness of this method for larger scale data is not clear from the paper.
-	Some important baselines and related work are not compared. For CLIP training, a standard data filtering method is CLIP-score (using a pretrained CLIP model to filter data). It will remove low image-text similarity samples effectively.  Other recent (and more advanced) data filtering papers for CLIP:
Data filtering networks. Neurips 23 workshops
Finetuned multimodal language models are high-quality image-text data filters. arXiv 24
CLIPLoss and Norm-Based Data Selection Methods for Multimodal Contrastive Learning. ArXiv 2024
They are not mentioned or compared in this work.
-	The authors claim that they are the first to comprehensively study the data efficiency problem within the context of contrastive pre-training (Line 100). It is an overclaim due to the numerous existing works (mentioned above).
-	Generally, the performance gap between SCAN and existing methods are small. In table2, the performance gap between SCAN and previous method Info-Batch is small (~0.5%). In table 3, SCAN and Info-Batch has similar performance on ImageNet zero-shot cls task.

**Questions:**

NA

---

### Official Review · Reviewer_2Wwx · 2024-11-03

**Soundness:** 3
**Presentation:** 3
**Contribution:** 3
**Rating:** 6
**Confidence:** 2

**Summary:**

The paper addresses the data efficiency problem in contrastive pre-training by introducing a novel dataset condensation method called SCAN. This approach aims to minimize the dataset size while maintaining performance, focusing on dynamic data selection rather than static dataset pruning. SCAN operates by iteratively pruning redundant and ill-matched data samples during training to optimize data use. Applied to two contrastive frameworks, CLIP and MoCo, SCAN shows only minor performance degradation despite using significantly reduced data. The approach outperforms static coreset selection methods, providing a new way to achieve efficient pre-training without excessive computational overhead or loss of model effectiveness.

**Strengths:**

1. SCAN’s dynamic pruning approach represents an advancement over traditional static pruning techniques, effectively enhancing data efficiency by continuously adjusting the dataset based on sample relevance throughout training.
2. The paper is well written and the proposed method is easy to follow.
3. The approach achieves comparable performance to full data training, even with up to 35% data pruning. This success indicates its potential as a resource-saving, efficient method for contrastive pre-training.

**Weaknesses:**

1. Incorporating the results without data pruning into Table 7 and Table 8 would enhance the clarity and provide a more intuitive presentation of the findings.
2. Interest in the extreme dataset pruning with the pruning ratio of 70%~90%. In my opinion, the performance of SCAN relies heavily on the careful tuning of pruning ratios, which requires additional experimentation to achieve optimal results across different datasets.

**Questions:**

Please see the weakness part.

---

### Official Review · Reviewer_aCqz · 2024-11-03

**Soundness:** 3
**Presentation:** 2
**Contribution:** 2
**Rating:** 5
**Confidence:** 4

**Summary:**

The paper proposes a data pruning method, resulting in improved training speed thanks to reduced training dataset size while negligible performance drop. The key features of the method are:
1. Dynamic. The method prunes training data dynamically and periodically.
2. Pruning metric. Based on training loss thresholding, the method prunes samples with very large and very small loses. The former are usually mis-matched samples, while the latter are usually redundant ones.

**Strengths:**

The idea is straightforward and demonstrates effectiveness by the experimental results.

**Weaknesses:**

1. Please improve the paper writing to avoid unnecessary confusion. (See "Questions")
    a. Please clearly state the experimental settings in each section. E.g., what are the settings (task & model) in Table 6?
    b. How are the total numbers of training steps determined for different settings? Are they are always same?
    c. Quantitatively, what is the overhead of pruning operations?
    d. Why did Info-Batch model on ViT-B/32 collapse in Table 1? And Swin-Base in Table 3?
2. I am concerned about the generalizability of the method. It has multiple heuristic parameters to tune. The selection of hyper-parameters is (highly) dependent on task (thus dataset, loss type), model, even batch size.
3. In Section 4.3 - Table 4, it might be better to show if only "SCAN (static)" contributes to the "competitive results", or both "SCAN (static)" and model ensemble (RN50 and ViT-S/16). To exclude the model ensemble effects, maybe just use a single model for static pruning.
4. In Section 4.3 - Table 4, does it mean static pruning method is even better (simpler, but comparable results) than the dynamic method based on results from table 4, especially RN 50 results? (ViT-B/32 is not a good comparison base since the pruning is using RN50 and ViT-S/16, which have domain gaps with ViT-B/32, as shown in Table 5.) This is somewhat aligned with results in Table 7: ill-matched pruning matters most, and ill-matched samples are almost always bad no matter what training epoch it is at.
5. Glad to see the ablation studies on pruning ratio in Sec 4.4. Maybe it's worth to expand the pruning ratio range. In a wider range, we perhaps will not see it's any close to a linear relationship assumed in Figure 3.
6. When ablating Mutation Epochs, can authors provide any interpretation for the results? Why not the more frequent the better?
7. Table 7 indicates ill-matched pruning is clearly more important than redundant pruning, but Section "Different Variants of the Same Pruning Ratio." finally chose even distribution based on Table 8.  Maybe it's worth doing more fine-grained ratio ablation.

**Questions:**

See above.

---

### Official Review · Reviewer_AWqX · 2024-11-04

**Soundness:** 3
**Presentation:** 3
**Contribution:** 3
**Rating:** 6
**Confidence:** 3

**Summary:**

This paper introduces SCAN, a dynamic bootstrapping dataset pruning method aimed at improving data efficiency in contrastive learning-based pre-training. SCAN utilizes a two-step approach—pruning data preparation and dataset mutation. To minimize training overhead, SCAN employs training loss as the data selection metric, effectively identifying valuable data while reducing dataset size. The method is versatile, working with both visual-only models and multi-modal models Experiments demonstrate that SCAN achieves comparable performance to models trained on the full dataset, with only marginal degradation at a pruning rate. Additionally, SCAN outperforms previous dataset pruning methods, such as SemDeDup, D-Pruning, and Info-Batch, validating its effectiveness

**Strengths:**

1. The target problem of data efficiency is important and has the potential for broad applicability.
2. The proposed method is simple yet highly effective. Its effectiveness is thoroughly validated through extensive experiments across various components, hyperparameters, datasets, and architectures. Additionally, the paper includes numerous ablation studies and analyses to support its claims.
3. The paper is well-structured, making it easy to follow.

**Weaknesses:**

The method appears quite straightforward, but both the approach and motivation seem effective. In my opinion, the experiments are sufficient to demonstrate its effectiveness.

**Questions:**

See the weakness section.

---

### Note · Authors · 2024-11-13

I have read and agree with the venue's withdrawal policy on behalf of myself and my co-authors.